# Cellular and Genetic Background of Osteosarcoma

Inga Urlić [1,*] , Marijana Šimić Jovičić [2] , Karla Ostojić [1] and Alan Ivković [3,4,5,*]

1   Department of Biology, Faculty of Science, University of Zagreb, 10000 Zagreb, Croatia;
    karla.ostojic@biol.pmf.hr
2   Department of Paediatric Orthopaedics, Children's Hospital Zagreb, 10000 Zagreb, Croatia;
    marijana.simic81@gmail.com
3   Department of Orthopaedics and Traumatology, University Hospital Sveti Duh, 10000 Zagreb, Croatia
4   School of Medicine, University of Zagreb, 10000 Zagreb, Croatia
5   Professional Study in Physiotherapy, University of Applied Health Sciences, 10000 Zagreb, Croatia
*   Correspondence: inga.urlic@biol.pmf.hr (I.U.); alan.ivkovic@gmail.com (A.I.)

**Abstract:** Osteosarcoma describes a tumor of mesenchymal origin with an annual incidence rate of four to five people per million. Even though chemotherapy treatment has shown success in non-metastatic osteosarcoma, metastatic disease still has a low survival rate of 20%. A targeted therapy approach is limited due to high heterogeneity of tumors, and different underlying mutations. In this review, we will summarize new advances obtained by new technologies, such as next generation sequencing and single-cell sequencing. These new techniques have enabled better assessment of cell populations within osteosarcoma, as well as an understanding of the molecular pathogenesis. We also discuss the presence and properties of osteosarcoma stem cells—the cell population within the tumor that is responsible for metastasis, recurrence, and drug resistance.

**Keywords:** osteosarcoma; cancer stem cells; driver mutations

## 1. Introduction

Osteosarcoma describes a malignant mesenchymal tumor whose cells produce osteoid, a non-mineralized bone matrix. According to the classification of tumors of soft tissues and bones by the World Health Organization (WHO), there are several types of osteosarcomas, of different malignancy grades. The subject of this review article is the conventional (classical) osteosarcoma (OS), which, according to the WHO, also includes secondary osteosarcomas. OS is the most common primary non-hematopoietic malignant bone tumor, with a worldwide incidence of 3.4 per million [1], which accounts for about 35% of all primary malignant bone tumors. It is the most common primary malignant mesenchymal tumor in children and adolescents, and in that age group, it accounts for about 80% of all malignant bone tumors. Over the age of fifty, OS accounts for about 50% of all primary malignant bone tumors, with a higher share of secondary OS, and is in second place in the overall share of primary malignant bone tumors in this age group [2]. This tumor has a highly heterogeneous genetic profile, but advances were made in understanding its biology and revealing genetic aberrations that define possible patient subgroups. These developments were made feasible by the emergence of thoroughly annotated tissue banks, along with the advancement and broader availability of technologies for detailed molecular profiling [3]. This review aims to gather current knowledge in the field of osteosarcoma molecular biology.

## 2. Cell Populations in Osteosarcoma

OSs are very complex tumors, because they contain heterogeneous sets of cells that differ in morphology, phenotype, gene expression, metabolism, immunogenicity, and proliferative and metastatic potential. Osteosarcomas grow in very complex and dynamic bone microenvironments, made up of bone cells (osteoclasts, osteoblasts, and osteocytes),

stromal cells (mesenchymal stem cells and fibroblasts), vascular cells (endothelial cells and pericytes), immune cells (macrophages and lymphocytes), and a mineralized extracellular matrix (ECM).

Precise intratumor cell populations could not be distinguished using conventional transcriptomics due to insufficient resolution of the bulk analysis. However, single-cell RNA sequencing (scRNA-seq) provided insight into the complex intratumor heterogeneity and tumor interaction with the microenvironment. Zhou et al. performed scRNA-seq on 7 primary, 2 recurrent, and 2 lung metastatic osteosarcomas and identified 11 cell clusters based on unbiased clustering of gene expression profiles and canonical markers [4]. Different cell clusters included osteoblastic cells, myeloid cells, osteoblastic proliferative cells, tumor-infiltrated lymphocytes (TILs), chondroblastic cells, endothelial cells, mesenchymal stem cells (MSCs), pericytes, fibroblasts, and myeloblasts [4,5]. In an analysis performed by Wu et al. [5], the main cell types were osteoblastic, myeloid, osteoblastic proliferative, and osteoclasts. However, the composition changed in metastatic OS, and included a large proportion of TILs, osteoblastic cells, and MSCs. In recurrent OS, there was a large proportion of chondroblastic cells. TILs were recruited to the tumor site, and several studies pointed out their importance for clinical outcome. The higher infiltration of immune cells was correlated with better overall survival and progression-free survival of OS patients [4].

A plethora of growth factors are released from the degraded bone matrix, as they establish a vicious cycle of bone remodeling processes. Pediatric osteosarcomas have a peak incidence around the time of the puberty, mostly located in the growth plate of the long bones. This suggests a connection between intensive bone growth and OS [6]. Bone remodeling relies on precise communication between various cell types crucial for bone homeostasis, including bone matrix-secreting osteoblasts and bone-degrading osteoclasts. MSCs are a part of the bone marrow niche, and they can differentiate into osteoblasts under the control of specific transcription factors. Growth factors, such as transforming growth factors (TGFs) and fibroblast growth factors (FGFs), must first engage a sophisticated signal transduction cascade in order to activate those transcription factors [7]. Most of these cytokines and growth factors are connected to the development of OS. The tumor microenvironment is rich in TGF-β, a factor that stimulates growth and metastasis. TGF-β-1 is mostly linked to the formation of OS during the growth of primary tumors or metastatic progression [8]. FGFs are crucial regulators of skeletal development, as FGF18 is required for the maturation of osteoblasts and FGF2 is essential for the growth and development of pre-osteoblasts [9]. Mostly, osteosarcomas are osteolytic, suggesting increased osteoclast activity [10]. Osteoclast activity is regulated by many cytokines and growth factors, such as macrophage-colony stimulating factor (M-CSF), insulin-like growth factors (IGFs), parathyroid hormone (PTH), RANK ligand (RANKL), and osteoprotegerin (OPG).

## 3. Cellular Theory of Osteosarcoma Tumorigenesis

### 3.1. Osteosarcoma Stem Cells

The tumorigenesis of OS is not fully explained; however, as for other malignant tumors, it is complemented by different theories, the latest being the theory about tumor stem cells. According to that theory, in the evolutionary progression of a malignant tumor or its recurrence after therapy, the main unit of selection is the cell, and, in particular, the one that has a high potential for a self-renewal—the cancer stem cell (CSC). The hypothesis of the CSC's existence was created as part of experiments on the transplantation of leukemic cells, and although it was published as a general feature of all malignant tumors, it is still disputed [11–13]. This theory also applies to OS stem cells, OS-CSCs, which is supported by clinical cases describing the appearance of metastases after more than 20 years of remission [14].

OS-CSCs theoretically arise either by transformation of undifferentiated MSCs, or by transformation of directed, progenitor cells of the osteoblastic lineage [15]. MSCs are non-hematopoietic precursor cells found in numerous tissues, including bone marrow, peripheral blood, placenta, umbilical cord, and adipose tissue, which show the ability

to self-renew and differentiate into various mesenchymal cell lines, such as osteocytes, chondrocytes, myocytes, and adipocytes, and have roles in wound healing [16]. According to this assumption, MSCs are recruited from the bone marrow (BM-MSCs) in response to genetic and epigenic changes and microenvironmental signals [17]. The histological type of mesenchymal tumors may be related to the type of oncogenic stress in MSCs. Thus, for example, it was found that a loss of *p53* expression and a simultaneous loss of expression of the *Rb* and *p53* genes led to the formation of leiomyosarcoma in vivo, regardless of the source of MSCs [16]. BM-MSCs *(Rb -/- p53 -/-)* differentiated into the osteogenic lineage and developed tumors in vivo with histological features consistent with OS [18,19]. A change in cell cycle regulators can also occur in osteoblast precursors, which indicates that they could also be a source of OS, orchestrated by signals from the tumor microenvironment and responses to genetic and epigenic changes within these cells [20].

OS-CSCs interact with local cells. Circulating normal MSCs that enter a tumor can support its growth, enhance invasiveness, and participate in the creation of metastatic niches in cooperation with tumor cells. Under the influence of MSCs, dedifferentiation of OS cells can occur so that they show stemness properties [21,22]. Among other ways, this intercellular communication occurs via vesicles. Researching the influence of the aforementioned vesicles secreted from OS cells, it was found that MSCs can internalize vesicles, after which such so-called tumor-directed MSCs play a role in the promotion of OS growth and metastasis [23]. Vesicles secreted from OS cells contain TGF-β (transforming growth factor beta), which is important for the development of the CSC's phenotype [24].

CSCs play a significant role in achieving tumor heterogeneity, by constantly enriching the tumor mass with new mutated cells and dominant subclones, and by regulating the microenvironment of tumor cells [25]. Generally, the presence of stem cells defines the microenvironment in which they reside, and the microenvironment influences the properties of stem cells. It consists of CSCs, fibroblasts, inflammatory cells, capillaries, extracellular matrix, and chemokines, and cytokines [26]. This microenvironment facilitates the entry of CSCs into a state of rest, retention of stemness properties, and self-renewal, and provides them physical support. It strengthens the resistance of CSCs to stresses, such as hypoxia, the immune system, antitumor drugs, and radiotherapy [27,28]. The tumor can colonize tissue locally and remotely, and CSCs play a key role in this process, as they contribute to the creation and maintenance of the tumor niche [29].

Fibroblasts associated with a tumor (cancer-associated fibroblasts, CAFs) are activated fibroblasts that share similarities with normal fibroblasts, and are stimulated by inflammatory conditions associated with tumor development. They represent a significant component of the stroma in tumor tissue, surround tumor cells, and play an important role in mechanical support, proliferation, survival, angiogenesis, metastasis, and immunogenicity [30]. Stem properties can be enhanced by conditioned medium from CAFs, mediated by the presence of paracrine-secreted molecules that suppress tumor differentiation, in which, for example, FGF (fibroblast growth factor) plays an important role [31]. Poor differentiation of the human OS cell line MG-63 could be associated with overproduction of basic FGF (bFGF) and bFGF receptors [32].

OS-CSCs may represent a separate therapeutic target that complements conventional treatment, so elucidating the characteristics and role of CSCs in OS may improve its treatment [33]. The necessity to understand the complex osteosarcoma microenvironment and CSC niche led to the development of new, better research models—tissue-engineered 3D models [34]. Wang et al. (2022) used 3D printing technology to fabricate a PLLA-based biomimetic scaffold resembling the physiochemical environment of cortical bones. A comparison of the transcriptome data revealed that 3D models can recapitulate OS properties better than patient biopsy samples and better than spheroid cultures. KEGG pathway analysis confirmed 20 enriched pathways, especially metabolic, MAPK, PI3K, and TGFbeta/SMAD pathways, in engineered tumors in comparison to spheroid cultures [35].

### 3.2. Phenotypic Characteristics of Osteosarcoma Stem Cells

Gibbs et al. were the first to prove the existence of OS-CSCs by identifying a subpopulation of cells that were able to form sarcospheres (Figure 1) in serum-free conditions in OS tissue samples obtained from patients [36].

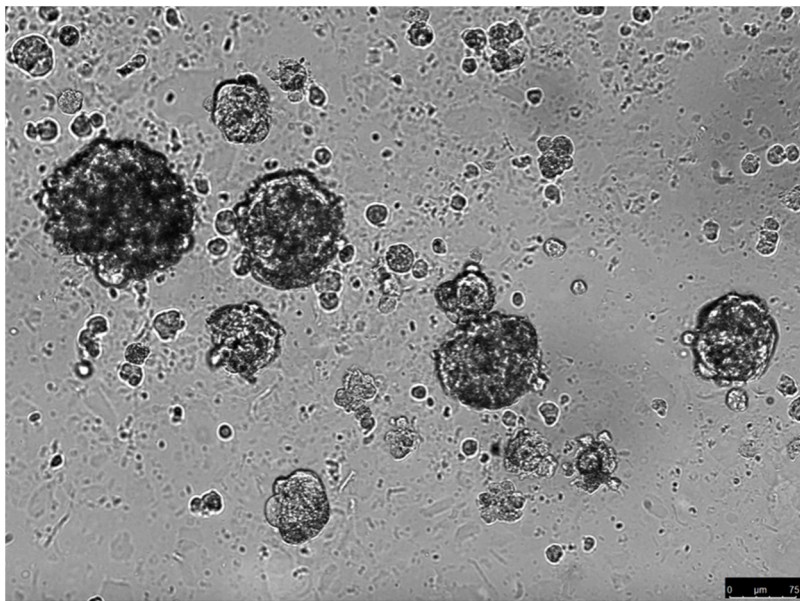

**Figure 1.** Sarcospheres growing in serum-free conditions (unpublished data). The spheres were derived from the parental cell culture of an osteosarcoma biopsy. The spheres were cultured in methyl-cellulose with serum-free medium for 14 days.

It is believed that one of the mechanisms by which CSCs achieve drug resistance is by increasing the level and activity of transporters responsible for the elimination of drugs from the cell, such as MDR/ABC (multidrug resistant transporters/ATP-binding cassette) [37]. This feature is used to identify CSCs by measuring the ability of the cell to release a dye that binds to DNA (Hoechst 33342 or Rhodamine 123) and sorting the cells by measuring fluorescence [38]. Cells that express a high level of the mentioned transporters emit color during the analysis and are also labeled as side populations in the literature [39]. The presence of a cell population that eliminates Hoechst 33342 dye has been demonstrated in OS cell lines and in human primary OS [40]. Martins-Neves et al. have identified such subpopulations of cells in the MNNG/HOS tumor line that showed resistance to antitumor drugs and radiotherapy [41].

OS-CSCs are phenotypically similar to stem cells due to the expression of genes for the Oct-3 and Oct-4 proteins (transcription factors that bind octamer 3 and octamer 4, respectively), which play a role in the self-renewal of undifferentiated embryonic stem cells and are often used as markers for undifferentiated cells. In addition to them, OS-CSCs also express the Nanog protein, encoded by the *Nanog* gene, which is a transcription factor that helps embryonic stem cells (ESCs) to maintain pluripotency by suppressing differentiation cell factors. Therefore, *Nanog* gene deletion triggers ESC differentiation. Basu Roy et al. found cells that have an increased ability to grow in non-adherent conditions by forming spheres, express the stem cell markers Sox2 (SRY-box transcription factor 2) and Sca-1 (stem cell antigen 1) in lineage and non-lineage OS samples [42]. Loss of *Sox2* gene function resulted in enhanced OS cell differentiation and decreased clonality, migration, and invasiveness [43].

In the last few decades, numerous studies have tried to identify specific markers and properties of OS-CSCs [44]. In addition to the expression of stem cell markers shared with ESCs, OS-CSCs have the ability to form cell spheres in vitro, which are highly tumorigenic in vivo and have a high level of aldehyde dehydrogenase-1 (ALDH-1) [45]. The ALDH-1

expression is associated with resistance to antitumor drugs and enhanced metastatic potential in tumor cells [46]. Adhikari et al. showed the presence of cells expressing CD117 (a protein that binds to stem cell factor, SCF) and Stro-1 (MSCs marker) in mouse and human OS cell lines. The double-positive cell fraction, compared to the double-negative cell fraction, showed a greater ability to form spheres, an increased resistance to doxorubicin treatment, greater metastatic potential, and overexpression of the ABCG2 drug efflux transporter (ATP-binding cassette super-family G member 2) [47]. CD44 is another marker of MSCs and plays a role in the migration and metastasis of OS-CSCs [48,49]. CD133, used as a marker for CSCs of various types of tumors, including OS, is strongly expressed in patients with lung metastases and a poor prognosis. Oct4+ OS cells that also expressed CD105 had a greater propensity to metastasize to the lungs [50,51]. CD105 is a cell membrane glycoprotein mainly expressed on endothelial cells and overexpressed in the tumor endothelium, and is a component of the receptor for TGF-β [52]. Furthermore, OS-CSCs express CD271, a low-affinity receptor for neural crest growth factor and a brain stem cell marker. CD271+ cells showed resistance to antitumor drugs and higher sphere-forming potential, as well as higher tumorigenic potential after injection into nude mice [53].

## 4. Driver Mutations and Epigenetic Changes of Osteosarcoma

### 4.1. Coding DNA Changes in Osteosarcoma

Osteosarcoma develops under the pressure of random mutational changes, and its growth is led by preferential clonal proliferation and epigenetic modifications [54]. Understanding the pathogenesis of osteosarcoma is complicated by the low tumor incidence and significant genetic differences between subtypes. For easier understanding, we will categorize OS into two groups based on their origin—hereditary syndromes, and sporadic OS that can be further categorized as pediatric OS or adult OS.

Several hereditary syndromes and genetic alterations have been linked to OS, such as Li–Fraumeni and retinoblastoma syndromes, as well as germline mutations in genes encoding DNA helicases of the RecQ family [55].

Li–Fraumeni syndrome (LFS), caused by a germline mutation in the *TP53* gene, is the syndrome that most frequently predisposes children to sarcomas. *TP53* encodes for p53, a master transcription factor, and about one-third of people with LFS develop OS. Therefore, it is no surprise that loss of the tumor suppressor function of p53 also happens in sporadic cases of OS [56]. Approximately 20% of OS cases have mutations in the *TP53* gene [55]. Protein p53 is involved in many antitumor cellular processes regulating cell growth, cell division, apoptosis, and DNA repair. Mutations in OS may result in loss of function or gain of function. The majority of mutations are missense mutations localized in the DNA binding domain. Additionally, 74% of OS cases have structural variations (SV) and somatic nucleotide variations (SNV). Among those, translocations with a breakpoint in the first intron result in the loss of *TP53* expression [57]. Other intron mutations result in splice site changes or a frameshift in sequence [58]. Amplification of *p53* main negative regulator MDM2 is also found in 10% of OS cases as a part of a larger amplicon containing CDK4, MDM2, and SAS [59]. In total, 32 variants of the *TP53* gene were found in tumor samples from 765 patients with LFS [60] as well as some heterozygous mutations of the *CHK2* gene [61].

Another tumor suppressor gene linked with OS is *Rb,* which codes for pRb. The carriers of a germline mutation (autosomal dominant) of *RB* have an increased risk of various neoplasms, especially OS. The incidence of OS in hereditary retinoblastoma is 400 times higher than in the general population [62]. The *Rb* gene is often altered in 70% of sporadic OS cases. It is affected by point mutations or structural variations. The very common finding is a loss of heterozygosity at 13q [63]. The Rb protein is an important guardian of cell cycle entry, and loss of its function leads to uncontrolled cell division. Other upstream and downstream components of the Rb pathway can also be affected by mutations in OS cases. pRb is regulated by phosphorylation via complexes of cyclin-dependent kinases and their coactivator cyclins. Phosphorylation of pRb activates cell

division. CDK4 and cyclin D1 are found upregulated in OS, while CDK inhibitors are commonly lost [59].

It has been shown that a loss of RecQ helicases, which unwind DNA prior to replication, also represents an increased risk of developing OS (Table 1). Germline mutations in the RecQ family of genes cause rare recessive autosomal cancer predisposition syndromes—Rothmund–Thompson II (RTS II), RAPALIDINO, and Bloom and Werner syndromes—which are associated with an increased frequency in OS development [55,64].

**Table 1.** Osteosarcoma-predisposing syndromes caused by mutations in the RecQ family gene.

| Syndrome | RecQ Family Gene |
|---|---|
| Rothmund–Thompson type II | *RECQ4*—ssDNA annealing activity |
| RAPALADINO | *RECQ4*—ssDNA annealing activity |
| Bloom | *BLM*—DNA helicase |
| Werner | *WRN*—DNA helicase |

Another syndrome, named Diamond–Blackfan anemia, is caused by a mutation in the gene for ribosomal protein, and is classified as ribosomopathy. Two studies reported an increased risk for OS in Diamond–Blackfan anemia patients [65], and mutations were found in the genes *RPS19, RPL5, RPL11, RPL35A, RPS24, RPS17, RPS7, RPS10,* and *RPS26.*

The *c-Myc* gene is mutated in more than 10% of OS cases. It is a known oncogene, mainly affecting cell proliferation and migration by activating the MAP kinase pathway [56]. A high expression of *Myc* correlates with metastatic disease and poor prognosis [55].

In adult OS cases, 14% of tumors present IGF1 receptor amplification [66]. In addition to previously described genes, sequencing data revealed other driver mutations of *Notch1, FOS, NF2, WIF1, BRCA2, APC, PTCH1,* and *PRKAR1A.* Additionally identified were synergistic genes including *Rb1, TWIST, PTEN,* and *JUN* [67].

Two very distinguished mutational processes are involved in osteosarcoma genesis, causing a heterogenous intratumoral profile of OS. The first one is complex chromosomal rearrangement, also known as chromothripsis, present in about 77% of OS cases [68]. It is a single catastrophic event that results in several genomic rearrangements in one or more chromosomes, including amplifications of *CDK4, MDM2, COPS3,* gains of *RICTOR, TERT,* and losses of *TP53* and *NF1* [69]. The second process is kataegis—an accumulation of mutations at localized regions (50–85% of cases) [70]. Those localized hypermutated regions are common in the regions of the *TP53* and *ATRX* genes [71].

### 4.2. Non-Coding RNA Changes in Osteosarcoma [55,56,60–71]

Aside from alterations in the protein-coding genes, it is evident that many non-coding RNA (ncRNA) genes are dysregulated in OS. ncRNAs are commonly used as tumor markers due to easy sample acquisition, and higher sensitivity and specificity than protein markers. Clinically relevant ncRNAs include microRNA (miRNA), long non-coding RNA (lncRNA), and circular RNA (circRNA). miRNAs are often 18–25 nucleotides in length and regulate gene expression targeting specific mRNAs via complementary binding with 3′UTR of mRNAs [55,56,60–71]. lncRNAs have over 200 nucleotides and exert their functions mainly via sponging miRNAs and targeting specific substrates.

lncRNAs act as miRNA sponges, preventing miRNA from acting on mRNA, therefore enhancing the translation of target mRNA (Figure 2). Interestingly, miRNA can also negatively regulate the expression of lncRNA. In the context of osteosarcoma, the affected signaling pathways are Notch, PI3K/AKT, JNK, and Wnt [72], important for cell proliferation, cell survival, self-renewal, and drug resistance. One example is lncRNA SNGH12 inhibiting miR-195-5p, which inhibits Notch2. Very similar is the case of lncRNA SNGH7 that inhibits miR-34a, inhibiting Notch1, BCL-2, Ki-67, and CDK6. As a consequence, in osteosarcoma, Notch1 or 2 signaling is upregulated, promoting tumorigenesis and metastases [73,74]. JNK and P13/AKT signaling pathways are induced in osteosarcoma cells via axes lncRNA

CAT104—miR-381—mRNA ZEB1 [75], lncRNA MEG3—miR 127—mRNA ZEB1 [76], and lncRNA ATB—miR-200—mRNA ZEB1 and ZEB2 [77]. Inhibition of apoptosis, and cell cycle arrest, is the result of axes lncRNA PVT1—miR-195, miR-497—BCL2, CCND1 and HK2 [78], lnc RNA SNHG20—miR-139—RUNX2 [79], lnc RNA TUG1—miR132-3p, and miR-212-3p—SOX4, FOXA1 [80].

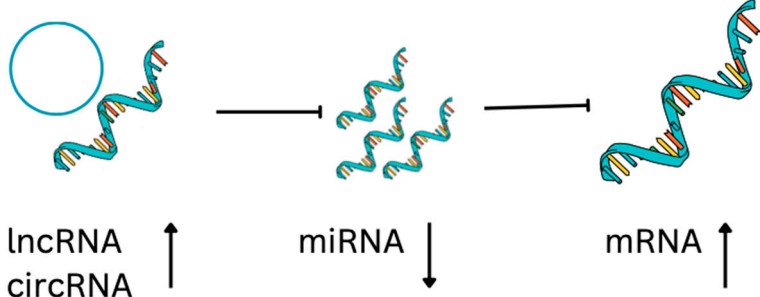

**Figure 2.** Regulatory ncRNA-mRNA axis. lncRNA and circRNA can suppress the miRNA that sponges mRNA. As a result of their presence, gene expression is upregulated. As a result of their absence, gene expression is downregulated.

Additionally, lncRNAs can suppress osteosarcoma proliferation and metastasis if the result of miRNA sponging is the upregulation of tumor suppressors. The examples affecting osteosarcoma cells are axes lncRNA CASC2—miR-181a—RASSF6 [81], PTEN and ATM, lncRNA-p21—miR-130b and PTEN [82], lncRNA NBAT1—miR-21—PTEN, PDCD4, TPM1, and RECK [83]. The situation between ncRNAs is even more complex because miRNA can negatively regulate lncRNA, and there is also reciprocal regulation between lncRNA and miRNA [72] (Table 2).

CircRNAs are ssRNAs that form closed loops and modulate gene expression, splicing, and sponging of miRNAs, and interact with RNA binding proteins [84] (Figure 2). For example, Qiu et al. [85] identified 15 downregulated circRNAs, 136 upregulated miRNAs, and 52 downregulated mRNAs from an OS database, in which 14 circRNAs, 24 miRNAs, and 52 mRNAs formed a network affecting common cancer-related pathways [84]. For example, CircTVF25 promotes cell proliferation and migration, upregulating cyclin D1, CDK6, MMP2, MMP9, and vimentin. circTCF25 reduces miR-206-inducing pathways MEK/ERK and AKT/mTOR [86]. CircPVT1 increases metastatic potential of osteosarcoma cells by promoting epithelial to mesenchymal transition. CircPVT1 reduces miR-205-5p activation of c-FLIP [87]. CircMMP9 is present in advanced osteosarcoma, it sponges miR-1265 activating chitinase-3-like protein 1 (CHI3L1) circMMP [88] (Table 3).

**Table 2.** lncRNA in osteosarcoma.

| Signaling Patway | lncRNAs | References |
|---|---|---|
| Notch | SNHG 12 | [75] |
| | SNGH 7 | [76] |
| PIK3/AKT | CCAT1 | [89] |
| | DANCR | [90] |
| | GAS5 | [91] |
| AKT | lncRNA-p21 | [84] |
| JNK and Wnt | CAT104 | [77] |
| | MEG3 | [78] |
| | ATB | [79] |
| p38/ERK/AKT and Wnt/β-catenin | C2dat1 | [92] |
| PI3K/AKT, JAK/STAT, and Notch | TUG1 | [93] |
| | NEAT1 | [94] |
| | PVT1 | [95] |
| | APTR | [88,96] |

**Table 3.** circRNA in osteosarcoma.

| Signaling Patway | circRNA | References |
|---|---|---|
| MEK/ERK and AKT/mTOR | circTCF25 | [88] |
| Akt/PI3K and Bcl-2 | circ-0001785 | [97] |
| Atk/PTEN | circ_ORC2 | [98] |
| MEK and ERK | circ-ITCH | [99] |
| VEGF, CDK4 | Circ_001621 | [100] |
| CREB3 | CircTADA2A | [101] |
| SOCS3 and STAT3 | Circ_ANKIB1 | [102] |
| β-catenin | circMYO10 | [103] |

*4.3. Osteosarcoma Extracellular Vesicles*

Extracellular vesicles (EV) are released from the cells in the extracellular matrix. Vesicles are surrounded by membranes and they carry cargo collected within the cell, enabling communication within the microenvironment, as well as distant communication. Their cargo is composed of proteins, DNA (mitochondrial and fragmented genomic), RNA (mRNA and ncRNA), lipids, and metabolites [104]. EV derived from osteosarcoma cells and mesenchymal stem cells (MSCs) from the tumor microenvironment are important for osteosarcoma progression, invasion, and metastasis. Via EV, OS cells can establish crosstalk with surrounding cells (osteoblasts, osteoclasts, osteocytes, MSCs, fibroblasts, immune cells, endothelial cells, and pericytes), modulate the epigenetic status of MSCs, and prepare a premetastatic niche. Many of the previously mentioned lncRNAs, miRNAs, and circRNAs are found in the EVs secreted by OS cells affecting angiogenesis, matrix remodeling, proliferation, and migration [105]. Examples of miRNAs in the EVs that support OS metastasis are miR-21 [106], miR-1307 [107], miR-675 [108], and miRNAs that change the OS microenvironment are miR-148-3p and miR-148-5p [109]. OS EVs also contain proteins that help the immune scape, such as PD-L1 [110]. EVs play an important role in the multidrug resistance of osteosarcoma. EVs transfer MDR-1 mRNA [111] and miR-135b [112], as well as circRNA_103801 [113], spreading resistance to chemotherapy-sensitive cells.

EVs can be easily collected from biological fluids (blood, urine, saliva, breast milk, amniotic fluid, cerebrospinal fluid, pleural effusion, and bronchoalveolar lavage fluid), and used for prognosis, diagnosis, and evaluation of the progression of osteosarcoma [114]. Potential biomarkers are proteins PD-L1 and N-cadherin [115], and ncRNAs miR-675, miR-25-3p, miR-21-5, miR143-3p, miR-101, and has-circ-103801 [104,110,116,117]. The interesting potential application of EVs is the delivery of therapeutics, especially for delivering ncRNA that will induce tumor suppressor activity, as well as reduce metastatic potential and cell proliferation.

## 5. Therapy and Resistance

Current treatment of osteosarcoma includes surgical resection and systemic chemotherapy, resulting in the 5-year event-free survival of approximately 70% of patients with localized disease. Unfortunately, overall survival rates are less than 20% in patients with metastatic and recurrent disease [118]. The outcome for patients has not changed for decades, despite many new targeted therapies for cancer.

One reason for this is the complexity of osteosarcoma molecular mechanisms and the influence of the microenvironment and immune system. Therefore, transcriptome and proteome analyses could point out differentially expressed genes in osteosarcoma and provide information about new potential therapeutic targets. These data were used by Chaiyawat et al. (2017) for generation of the list of therapeutic targets for already-approved drugs. Upregulated proteins with FDA-approved antineoplastic drugs were categorized as epigenetic regulators (DNMT1, HDAC1, and HDAC2), kinases (ERBB2,

FGFR1, KIT, MET, MTOR, and PDGFRalpha), proteasomes (PSMC5 and PSMC6) and other drugs (GSR and PARP1). The analysis was also performed in order to identify proteins that can be targeted with non-antineoplastic drugs approved for other diseases. Among immunosuppressants for rheumatic arthritis, leflunomide, the inhibitor of DHODH, can be repurposed for osteosarcoma. Digoxin, a drug in the group of cardiac glycosides, can also be repurposed for osteosarcoma [119]. Additionally, patient-derived 2D and 3D cultures of giant cell tumors of bone and desmoplastic fibroma showed an upregulation of genes involved in bone vicious-cycle-related genes (*RANK-L, RANK, OPN, CXCR4, RUNX2, FLT1*), suggesting the therapeutic combination of denosumab and lenvatinib. The clinical data confirmed the usefulness of molecular and pharmacological results [120].

The second reason for poor outcome is that an osteosarcoma is considered a drug-resistant tumor. Chemotherapy resistance can be linked to drug import and efflux, intracellular detoxification, resistance to apoptosis, DNA damage repair, tumor microenvironment, immunity, and aquired mutations of the drug target. In order to achieve a better drug response, it is necessary to unravel the underlying mechanisms of osteosarcoma resistance to treatment. The main causes for methotrexat resistance are inefficient drug delivery due to poor vascularisation, an acidic tumor microenvironment that inactivates the drug [121], and impaired cellular influx [122]. Drug efflux is mediated by a membrane-bound pump encoded by the *MDR1* gene, and doxorubicin-resistant cell lines strongly induce its expression [123]. The cytotoxic effect of doxorubicin is based on its DNA intercalation and binding to topoisomerase, leading to apoptosis. The topoisomerase status may be a progonostic factor in survival and response to doxorubicin [6,124]. Alterations in p53 activity and apoptotic pathways also contribute to therapy resistance. The nature of the p53 aberrations is very heterogenic in osteosarcoma patients, providing controversial data regarding its contribution to therapy resistance. However, some individuals can benefit from p53 reactivation via small molecule RITA [125]. Osteosarcoma cells upregulate DNA repair pathways to fight direct and indirect DNA damage caused by cisplatin, ifosfamide, and doxorubicin. A pathway involved in the repair of DNA damage linked to chemotherapy resistance is nucleotide excision repair (NER). Upregulation of the involved protein ERCC1 and polymorphisms of ERCC2 are markers of poor prognosis [126].

## 6. Conclusions

Osteosarcomas exert high inter- and intratumor heterogeneity. A cellular background of the disease reveals that tumors consist of many cell subpopulations, including predominantly cells of mesenchymal origin, osteoclasts, and immune cells. Among those, there are populations of osteosarcoma cancer stem cells that are responsible for drug resistance, disease recurrence, and metastases. A genetic background reveals that osteosarcoma has very variable genetic alterations, but can be grouped by hereditary and sporadic subtypes. Since there are two peaks of incidence, the sporadic type is divided into pediatric- and adult-onset cases. Driver mutations generally include dysregulation of cell division, inactivation of *p53* and *pRb*, and interference with DNA repair mechanisms. Dysregulation of signaling does not rely only on mutations, but also on ncRNA networks. ncRNA networks include the interactions between circRNAs, lncRNAs, and microRNAs that modulate the mRNA levels of oncogenes and tumor suppressors. Tumor cells also educate surrounding cells, changing local and distant environments by secretion of exosomes packed with proteins, as well as coding and non-coding RNA molecules. The proinflammatory microenvironment supports progression via cytokines and growth factors secreted from immune cells, osteoclasts, and tumor stroma.

Altogether, complexity and heterogeneity of the osteosarcoma system are the key reasons why targeted therapies have limited success and chemotherapy still dominates in OS treatment. Special focus should be put into combination therapies that target proliferating cells and cancer stem cells, change the microenvironment, improve the immune response, and fight drug resistance.

**Funding:** This research was funded by the Croatian Science Foundation, grant numbers IP-2018-01-7590 and IP-2020-02-9559.

**Data Availability Statement:** Not applicable.

**Conflicts of Interest:** The authors declare no conflict of interest.

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
