# Peer review of "Cellular and Genetic Background of Osteosarcoma"

_cimb, doi:10.3390/cimb45050276_

Round 1

Reviewer 1 Report

Comments and Suggestions for Authors

The concept of this review paper is about cellular and molecular components of the classical osteosarcoma. It is very interesting topic and authors reviewed some of the best research papers in the field. However, it needs to be written in a more direct language and would be better to classify all the aspects. 

Also, that would be great to use a table to summarize the information. I think this is potentially a great review paper but just need some more edits and modifications.  

Reviewer 2 Report

Comments and Suggestions for Authors

The authors of the present work described the last advances obtained with the new genomic and preclinical approaches focusing the attention on the implication of OS stem cells in the different tumor processes.

The manuscript looks like well written and organized. The authors have presented an interesting topic in the field of OS. The paper should be considered after major revisions.

1.       The authors should provide more information about the therapeutic approaches for OS patients, in order to introduce the topic of drug resistance described in the text;

2.       Well-defined preclinical models are needed to better reproduce the OS biochemical processes. Commercial cell lines cultured on common monolayer supports are in vitro systems not able to mimic the microenvironment of cancer diseases. 3D models represent a valuable research resource that allow reproducing the different stemness and biological behaviors of OS tumors. The authors described the results obtained with spheroids and organoids. These systems are not able to reproduce the 3D architectures of tumor arised in the bone microenvironment. For this reason, the authors should underline the last innovations in the field of 3D matrix-based preclinical models for the study of phenotypes, pathogenesis and drug sensibility. The following references should be included in the manuscript: “A 3D-printed scaffold-based osteosarcoma model allows to investigate tumor phenotypes and pathogenesis in an in vitro bone-mimicking niche. doi: 10.1016/j.mtbio.2022.100295”,  “A Rationale for the Activity of Bone Target Therapy and Tyrosine Kinase Inhibitor Combination in Giant Cell Tumor of Bone and Desmoplastic Fibroma: Translational Evidences. doi: 10.3390/biomedicines10020372” and “Scaffold-based 3D cellular models mimicking the heterogeneity of osteosarcoma stem cell niche. doi: 10.1038/s41598-020-79448-y”.

Reviewer 3 Report

Comments and Suggestions for Authors

In this review article Urlic et al. introduce the molecular aspects of osteosarcoma. This review is unique and informative. However, there are lack of appropriate citations in this paper, and it cannot be denied that the authors may have written the contents that are not based on evidence. Proper literature should be cited at least page 3, lines 115, 125, 127, page 6, line 256, and page 7, line 321.

Reviewer 4 Report

Comments and Suggestions for Authors

The manuscript entitled "Cellular and genetic background of osteosarcoma" by Inga Urlic and colleagues is potentially of great interest to a general audience in the scientific community. Nevertheless, it has relevant flaws that need to be overcome before publication

At the moment, the manuscript is little more than a list. The various topics are poorly discussed as well as the challenges to be overcome for the diagnosis, as well as the clinical and therapeutic management of osteosarcoma. A deep reflection on the limits of current knowledge and on how to overcome them is missing.

The multi-omics approaches in OS (and in particular proteomics) should be discussed.

Reviewer 5 Report

Comments and Suggestions for Authors

This is a good review in assessing the cellular and genetic background of osteosarcoma. The review is well structured and detailed in many aspects identifying many critical aspects in the  genetics and cellular components in the development of OS disease.

There are some parts of the review the authors should add more detail and there are number grammatical errors that the authors should correct.

In the abstract the authors state an annual incidence of 4-5 per million cases. Do they mean globally or is this a specific geographical population? Please expand on the data.

Abstract “Even though, a chemotherapy treatment has success in non-metastatic osteosarcoma, metastatic disease still has low survival rate of 20%”  should be …still has a low…

Line 67-68  MSCs are a part of bone marrow niche. Should be  …are part of the bone marrow niche.

Line 73  TFG-1 Should this be TGFb-1?

Line 122 “This microenvironment facilitates the entry of CSCs into a state of rest, retention of stemness properties, self-renewal and provides them physical support.” In regard to this microenvironment why is the development of osteosarcoma occurring in very similar specific sites in bones. Why not other regions of the skeleton?

Figure 1 Are these the authors images or are they from Gibbs et al? If from the Gibbs publication then it is published data. Could the authors add more info on these sarcospheres. How long to culture, how they are cultured and patient specifics. Very interesting work.

Line 257 “…higher sensitivity and specificity then protein.”  Should be “than protein”

Figure 2. Add more information in the legend to give the figure more context. This figure is not referenced or discussed in the text.

Line 305 Should be “mesenchymal stem cells (MSCs) from the tumor microenvironment a…”

Line 307 Which bone cells establish cross talk? Osteoclasts, osteoblasts, osteocytes or all of them?

Line 316 Can the authors specify the body fluids the EVs can be collected from? Are there preferred fluid fractions to collect these EVs in OS patients?

Line 327 should be “there is a population…” or “there are populations of”

Line 330  “subtypes”

Round 2

Reviewer 1 Report

Comments and Suggestions for Authors

Well done! The modifications made the review paper much insightful and easy to read. 

Reviewer 2 Report

Comments and Suggestions for Authors

The manuscript in the present form is acceptable for the pubblication 

Reviewer 4 Report

Comments and Suggestions for Authors

The manuscript has been notably improved after revision, and it is now acceptable for publication 

Reviewer 5 Report

Comments and Suggestions for Authors

The authors have addressed my comments satisfactorily